# Mitral Valve Leaflets Segmentation in Echocardiography using Convolutional Neural Networks

**Eva Costa**
Neadvance, Machine Vision, SA
Braga, Portugal
ecosta@neadvance.com

**Nelson Martins**
Neadvance, Machine Vision, SA
Braga, Portugal
nmartins@neadvance.com

**Malik Saad Sultan**
Instituto de Telecomunicações
FCUP*, Porto, Portugal
engr.saadsultan@gmail.com

**Diana Veiga**
Neadvance, Machine Vision, SA
Braga, Portugal
dveiga@neadvance.com

**Manuel Ferreira**
Neadvance, Machine Vision, SA
Braga, Portugal
mferreira@neadvance.com

**Sandra Mattos**
UCMF [†]
Recife PE, Brazil
ssmattos@gmail.com

**Miguel Coimbra**
Instituto de Telecomunicações
FCUP*, Porto, Portugal
mcoimbra@dcc.fc.up.pt

## Abstract

Rheumatic heart disease remains a major burden in the developing countries. The World Heart Federation proposed guidelines for the echocardiographic detection of the disease, in which the mitral leaflets' morphology assessment is a key indicator. The drawback is that these guidelines are dependent on the clinician experience. To overcome this limitation, we propose an automatic segmentation of the mitral leaflets using a new method based on convolutional neural network, specifically the $UNet$ architecture. The results indicate a median DICE coefficient of 0.74 in $PLAX$ and 0.79 in $A4C$ for the anterior mitral leaflet segmentation, while median DICE of 0.60 in $PLAX$ and 0.69 $A4C$ are met for the posterior leaflet. A visual evaluation of this segmentation approach by two cardiologists is in line with the numerical results. The false detection due to overestimation and artifacts remains an issue to be addressed in the future.

## 1 Introduction

### 1.1 Motivation

Rheumatic heart disease (RHD) is a preventable chronic sequel of acute rheumatic fever (ARF), an autoimmune response to group A streptococcal infection. Although being almost eradicated in high-income countries, it remains a major burden in the developing countries, where it causes most of the cardiovascular mortality and morbidity in the young [1]. RHD can be definite (clinically diagnosed) or borderline/sub-clinical (detected only by echocardiography). In a recent prevalence study, the RHD was followed globally over a period of 25-years, [2], and it was estimated that in

---

*Faculdade de Ciências, Departamento de Ciências de Computadores, Universidade do Porto
†Unidade de Cardiologia e Medicina Fetal, Real Hospital Português em Pernambuco

1st Conference on Medical Imaging with Deep Learning (MIDL 2018), Amsterdam, The Netherlands.

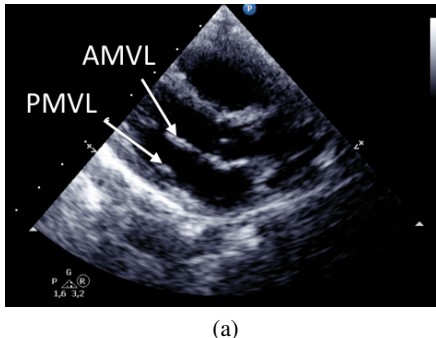 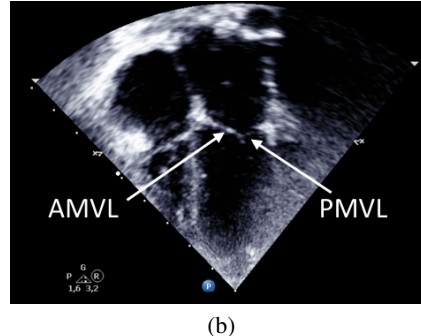

(a)  (b)

Figure 1: Brightness mode echocardiography. (a) parasternal long-axis view. (b) apical four-chamber view.

2015 alone, there were 33.4 million cases of RHD, 10.5 million disability-adjusted life-years related to RHD and 319400 RHD-related deaths. However, the estimations may fall short due to missing data in some regions of the globe, misidentification in the causes of death and due to sub-clinical RHD not being included in the prevalence study. This last aspect should not be disregarded since screening studies [3] point that for each case of clinical RHD, 3 to 10 cases of sub-clinical disease exist. It is important to note that even though sub-clinical cases may not develop into definite RHD, it is at this stage that the treatment is most effective with milder health repercussions.

With the advent of portable echocardiography and the increasing detection rates of sub-clinical RHD, an evidence-based set of guidelines was defined by the World Heart Federation (WHF) for the echocardiographic assessment of RHD [4]. RHD mostly affects heart valves, especially the mitral valve and, therefore, the WHF echocardiographic criteria are generally based on the morphology and functionality of this valve. The mitral leaflets' morphology and mobility is assessed through brightness mode echocardiography, usually in the parasternal long-axis view ($PLAX$), and in some cases using the apical four-chamber view ($A4C$). These echo views are shown in Fig. 1, with the anterior mitral valve leaflet depicted as AMVL and the posterior mitral valve leaflet as PMVL. Morphological assessment is usually done for AMVL instead of PMVL, solely because higher inter-observer agreement is met [4]. Clinical observation suggests the tip of the leaflet is the most commonly part to be affected [5]. Echocardiography assessment requires highly experienced operators, which is a scarce resource in developing countries. The use of image processing tools has the potential to reduce the operator dependency in screening settings, reduce the subjectivity and, in this way, improve the diagnosis.

## 1.2   State of the art

Ultrasound images are affected by several acquisition artifacts such as attenuation, speckle, shadows and signal dropout. Apart from that, the quality of the acquisition is strongly dependent on the human operator and the machine settings. The intensity and texture differences between structures and the contrast between structures and blood pool are low. These conditions raise several problems for classic image processing methods [6]. In [7], the authors proposed a combination of active contours and optical flow for the AMVL segmentation. The algorithm is semi-automatic and fails when the leaflet's displacement between frames is large and irregular. Another semi-automatic approach for the AMVL segmentation was proposed in [8]. Two connected active contours identify the cardiac muscle and the leaflet. The manual initialization and the parameters selection highly affect the method's performance. In [9], the authors propose a semi-automatic segmentation strategy, with a single point input from the user. The input point defines a set of scanning lines for a virtual motion-mode (M-mode) reconstruction. The posterior aorta wall's motion pattern is obtained by applying open-ended active contours to the virtual M-mode, using prior knowledge to establish constraints. The pattern provides a seed for each frame to segment the AMVL with localizing region-based active contours. Although it delivers the middle part of the leaflet, it sometimes fails to segment its tip, which is the most relevant part of the structure diagnosis-wise. An approach based on outlier detection in low-rank matrix was proposed in [10]. The authors aim to overcome the shortfalls of the

previous methods, with a fully automatic unsupervised method. However, this solution still requires an extensive parameter fine-tuning and cropping of the images around the region of interest. Also, the method does not discriminate between AMVL and PMVL. In [11] the authors claim to prevent tracking drifts caused by motion ambiguities by constraining the outlier pursuit, and refining the segmentation with region-scalable active contours. Significant parameter fine-tuning remains as a drawback.

Most of the literature approaches are highly sensitive to initialization, image quality and acquisition parameters. None of them segments both leaflets consistently. The complexity of the problem calls for supervised learning methods such as Convolutional Neural Networks ($CNN$). $CNNs$ have become the state of the art solution for image recognition problems, even outperforming human operators in some tasks. This approach will shift the burden of manual input from the final user to the training phase, and also will not rely on hand-crafted image features, making segmentation a fully automatic and robust process. To our knowledge there are no works in the literature on semantic segmentation of the mitral leaflets using $CNNs$. In [12], a partial segmentation of the mitral leaflets was needed to segment the left ventricle. The authors propose a network for patches' classification and then a second network for segmentation of the ventricle. However, they were not able to detect the contours of fast moving structures such as the mitral leaflets. The amount of data, and the respective manual annotations required for training a $CNN$ is a major point to take into account. The $UNet$ architecture, proposed in [13], claims to produce accurate results, with a small number of observations. This trait of the $UNet$ makes it an interesting contender for application in the present work, since the available dataset is also limited. The architecture allows for a multi-scale representation, with coarser information being collected in the bottom layers and finer information at the top ones. The architecture is composed by two paths: one of contraction, with convolutional layers and another of expansion with deconvolution layers. The paths are connected by skip layers before each max-pooling operation. This ensures that both local and global information is captured.

## 2  Proposed Work

In this work the $UNet$ will be used for the mitral leaflets' segmentation in the $A4C$ and $PLAX$ views. Each view produces distinct representations of the heart structures, thus, the model's development will be adapted for each one separately.

### 2.1  The $UNet$ model

The most favourable aspects of the $UNet$ architecture are that it does not require a large training set, and that only the image is needed as input. The least favourable trait of the $UNet$ is transversal to all $CNN$ architectures: the parameterization of the network requires a training phase. Depending on the complexity of the network, the training phase may require high computational power and time.

#### 2.1.1  Model implementation

The $UNet$ architecture was recursively implemented in TensorFlow's [14] front-end TFlearn [15] (Python), allowing expansion of the depth $D$ of the architecture (number of steps on the paths). In Fig. 2, the implemented $UNet$ architecture is shown.

Taking into account the specificity of the problem, some simplifications and changes were made to the $UNet$ model proposed in [13].

The first architectural change is the use of zero padding in the convolutions instead of the valid values. The authors of [13] proposed the use of the valid values with a mirror padding pre-processing, so the final outputs have the same spatial dimensions as the original input image. They argue that this accelerates the training, however this was not observed during preliminary tests and therefore, it was decided to use zero padding in all convolutions.

The second change is the use of batch normalization layers before the concatenation steps. This adds a regularization effect by ensuring that the concatenated feature maps have the same order of magnitude.

Since the present work is focused on the evaluation of the potential of $DNN$ architectures in segmenting the mitral leaflets', no extensive studies were made for the hyper-parameter optimization.

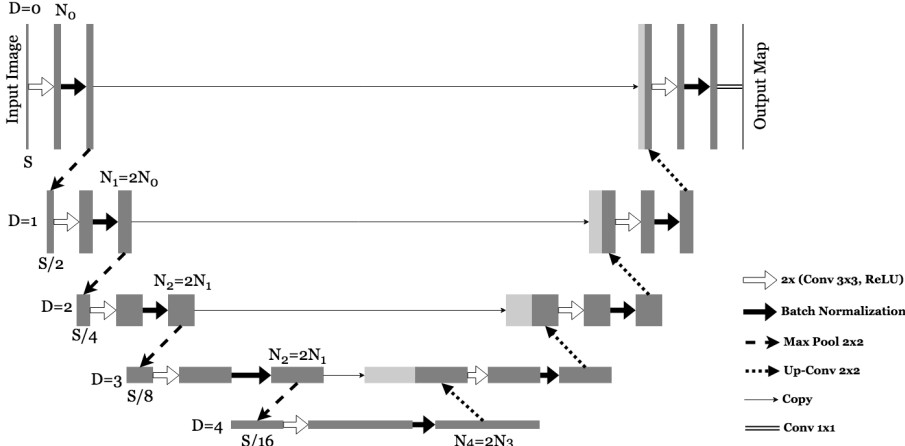

Figure 2: Proposed $UNet$ model, adapted from [13]. Normalizing layers were added before every $max - pool$ operation; $same$ padding in all $Conv$ layers, instead of $valid$. $N_x$ stands for the number of filters, $D$ for depth and $S$ for the spatial size of the feature map.

Most of the parameters were empirically chosen and maintained unchanged throughout the training stages: the *learning rate* was set to 0.001, the optimization algorithm was the $Adam$, a *batch size* of 4 was set, as loss function the *mean square difference* was selected, and as activation function the Sigmoid was used with threshold of 0.5.

The samples were randomly shuffled and divided into a training and validation set (75% for training and 25% for validation). Due to memory management and taking into account the down-sampling process, samples' dimensions were set to a 416 pixel height and 512 pixel width.

To achieve more accurate results, the architecture's depth $D$ and the number of incoming neurons $N_0$ were subject to a greed search optimization. Preliminary tests have shown that lower depths result in less accurate results.

## 3   Materials and Methods

The datasets used for this work contain videos from different patients, as described in Table 1.

Table 1: Distribution of the datasets for Training and Validation, Test and Application phases.

| Echo view | Train and Validation | Test | Application |
|-----------|----------------------|------|-------------|
| $PLAX$ | 21 videos (2163 frames) | 6 videos (520 frames) | 23 videos |
| $A4C$ | 22 videos (2400 frames) | 6 videos (526 frames) | 23 videos |

At the application phase, the tested $UNet$ models were applied to the dataset for clinical assessment. Two cardiologists were asked to evaluate the segmentation quality based on 6 parameters for the two views: overall detection of the each leaflet' tip pixels, overall estimate of each leaflet' thickness, amount of false positives and repeatability of the segmentation quality along the video. Each case was graded with scores of 0, 1 and 2 (0 connotes failure and 2 success).

The echocardiography sequences were acquired during the *Heart Caravan* of 2016, a health care provision initiative which took place in the State of Paraíba, Brazil. All images were acquired using a *Vivid I*, by GE and/or a *CX-50*, by Philips and from children with ages between 4 and 16 years old. The sets used for training, validation and test were manually annotated in each frame as depicted in Fig. 3.These annotations of the mitral leaflets (AMVL and PMVL) were made by an experienced user and validated by two pediatric cardiologists.

For the models' performance assessment during parameterization, the dice similarity coefficient ($DICE$) was used. This metric measures the similarity between the model's prediction and the manual annotation. For the evaluation of the $UNet$'s segmentation results, the $DICE$, precision

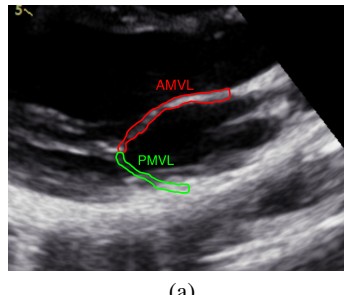 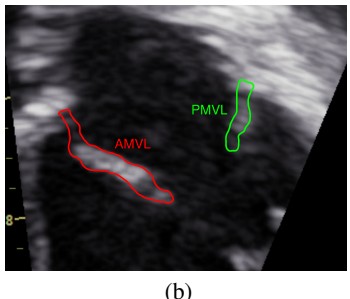

(a)                               (b)

Figure 3: Manual annotations of the AMVL, PMVL and CT. (a) parasternal long-axis view. (b) apical four-chamber view.

and recall were used. For the assessment of inter-rater agreement on the evaluation of the results in clinical context, the Bennett's $Sscore$ [16] was applied. The $Sscore$ estimates the agreement assuming that the likelihood of random agreement (both rater agree, when both select a category randomly) is solely dependent on the number of categories. For $q$ categories, $r$ raters and $n$ rated items, the $Sscore$ is defined as:

$$Sscore = \frac{p_o - p_c}{1 - p_c}, with:$$ (1)

$$p_c = \frac{1}{q^2} \sum_{k,l} w_{kl}, \qquad p_o = \frac{1}{n} \sum_{i=1}^{n} \sum_{k=1}^{q} \frac{r_{ik}(r_{ik}^* - 1)}{r_i(r_i - 1)}, \qquad r_{ik}^* \sum_{l=1}^{q} w_{kl} r_{il}$$ (2)

where, $r_{il}$ and $r_{ik}$ are the number of raters assigning item $i$ to category $l$ or $k$, respectively. Since in our case the categorization is a grading process, ordinal weighting was applied, thus when the two raters agree total credit is given ($w_{kl} = 1$), when raters disagree by choosing immediate neighbor categories partial credit is given ($w_{kl} = 0.33$), and when raters disagree by choosing non-neighbor categories no credit is given ($w_{kl} = 0$).

The $UNet$ models were trained in a Desktop PC with Intel Core i7 Processor (3.40 GHz), 16 GB RAM, and NVIDIA GeForce GTX 970 GPU with 4 GB RAM.

## 4  Results and Discussion

In this section we elaborate on the results gathered in this work, and on their evaluation. The following subsection covers the results of the $D$ and $N_0$ parameters grid search, the results on the test dataset and the results of the clinical evaluation of the proposed method.

### 4.1  Parameterization

The parameterization of the architecture's depth $D$ and number of incoming neurons $N_0$ was made for each echo view by evaluating the $DICE$ in the validation stage. The training stages were all stopped at 30 epochs and the model with higher $DICE$ in validation was saved. The highest result was always found before the $30^{th}$ epoch. Depths higher than 5 resulted in GPU memory overflow and, because of that, only depths of 4 and 5 were tested. The number of incoming neurons $N_0$ was studied in the range from 4 to 32 with base 2 steps.

#### 4.1.1  Parasternal long-axis

The results obtained in the validation stage are summarized in Table 2.

The highest $DICE$ was obtained for $D = 5$ and $N_0 = 8$. Even though the model lies in the border of $D$, further exploration was not made due to GPU overflow for depths higher than 5.

Table 2: Mean DICE Coefficient results in the validation set. In bold is the highest $DICE$.

| $D$ \ $N_0$ | 4 | 8 | 16 |
|---|---|---|---|
| 4 | <0.710 | 0.762 | 0.770 |
| 5 | <0.710 | **0.791** | 0.786 |

### 4.1.2 Apical four-chamber

The results obtained in the validation stage are summarized in Table 3.

Table 3: Mean DICE Coefficient results in the validation set. In bold is the highest $DICE$.

| $D$ \ $N_0$ | 4 | 8 | 16 | 32 |
|---|---|---|---|---|
| 4 | <0.710 | 0.757 | 0.757 | 0.760 |
| 5 | <0.710 | 0.756 | 0.762 | **0.771** |

Concerning $N_0$, contrary to what happened with the PLAX view, it was verified the results improved with higher values. Thus, the search was expanded. It was not possible to test $N_0 = 64$ due to hardware resource exhaustion. The model with highest $DICE$ has $D = 5$ and $N_0 = 32$.

### 4.2 Results on test dataset

In this section the $UNet$ segmentation quality is evaluated in the test set. From the 526 $A4C$ frames, 3 were excluded due to motion artifacts or probe mispositioning, which impeded the user from annotating the structures. The same happened with 12 of the 520 $PLAX$ frames.

In Fig. 4 the evaluation metrics' distributions are shown. An immediate assertion to be made is that the results are better in the AMVL than in the PMVL segmentation, in both views. This is in line with what happens with human observers, who have higher inter-observer agreement for the AMVL. Concerning the AMVL segmentation, the $DICE$ values are above 0.5 in $PLAX$ and above 0.6 in $A4C$, with a median of 0.742 in $PLAX$ and 0.795 in $A4C$. High recall values (0.903 median in $PLAX$ and 0.927 in $A4C$) indicate that most of the leaflets' pixels were correctly detected as such, so false rejection is not a significant issue. On the other hand, precision presents lower scores (0.688 in $PLAX$ and 0.710 in $A4C$), which might indicate that some false detection is happening. Post processing techniques may have a positive effect removing false positives. In what concerns PMVL segmentation, the same trends obtained in the AMVL are observed, yet the metrics present wider distributions. This denotes for higher variability in the results, with more false rejection and false detection. In $PLAX$, median values are $DICE$ of 0.600, recall of 0.787 and precision of 0.512. In $A4C$, median values are $DICE$ of 0.690, recall of 0.817 and precision of 0.615. Examples of the obtained segmentation results are shown in Fig. 5. The best $(a, d)$ and worst $(b, e)$ results are displayed for PLAX and A4C. This selection takes into account the average $DICE$ of the two classes (AMVL and PMVL). The best average result for PLAX is 0.848 and the worst is 0.354. The best average result for A4C is 0.869 and the worst is 0.260. Two examples of false detection errors are also shown: $(c)$ demonstrates overestimation of the structures' borders and $(f)$ demonstrates the presence of false positives due to reflection artifacts on the US response. In some cases, overestimation may not be a significant error, since some limits of the leaflets (AMVL - posterior wall of the Aorta boundary and PMVL - left atrium wall boundary) are almost arbitrarily chosen when manually annotating. These frames present high recall values, while precision is low.

### 4.3 Results on application dataset

The clinical evaluation results of the application dataset are summarized in Table 4.

The confusion matrices in Table 4 show that both raters assigned score 2 more often than 1, and 1 more often than 0. The pooled $Sscore$ is 0.781, which means a substantial agreement between raters, which reinforces the assigned scores. From all the evaluated parameters, the amount of false positives

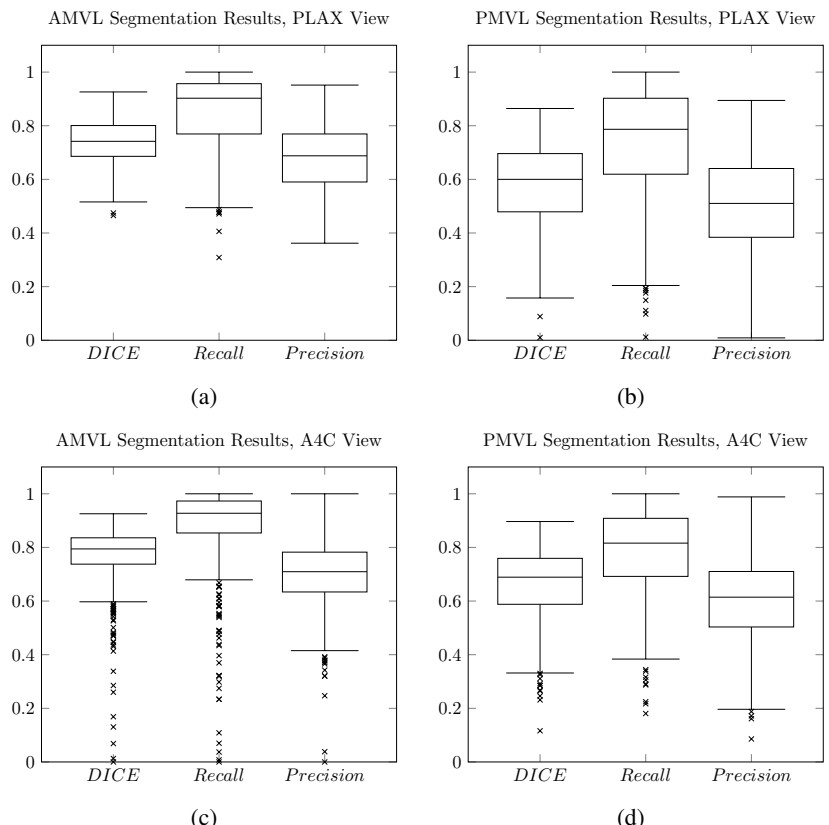

Figure 4: Boxplot of the evaluation metrics (DICE, recall and precision) obtained from the test images using the $UNet$ method. The $\times$ are the outliers. (a) AMVL segmentation in A4C view. (b) PMVL segmentation in A4C view. (c) AMVL segmentation in PLAX view. (d) PMVL segmentation in PLAX view.

Table 4: Results of the clinical evaluation of the segmentations' quality by two raters ($R1$ and $R2$). P1: AMVL tip detection; P2: AMVL thickness; P3: PMVL tip detection; P4: PMVL thickness; P5: amount of false positives; P6: repeatability along the video. S2, S1 and S0 stands for the grading scores.

| | | P1 | | | P2 | | | P3 | | | P4 | | | P5 | | | P6 | | |
| | | $S2$ | $S1$ | $S0$ | $S2$ | $S1$ | $S0$ | $S2$ | $S1$ | $S0$ | $S2$ | $S1$ | $S0$ | $S2$ | $S1$ | $S0$ | $S2$ | $S1$ | $S0$ |
|---|---|---|---|---|---|---|---|---|---|---|---|---|---|---|---|---|---|---|---|
| $R1$ | $S2$ | 20 | 1 | 0 | 19 | 3 | 0 | 21 | 0 | 0 | 19 | 1 | 0 | 6 | 3 | 1 | 18 | 2 | 0 |
| | $S1$ | 1 | 1 | 0 | 0 | 0 | 1 | 0 | 1 | 1 | 1 | 1 | 1 | 4 | 7 | 0 | 2 | 0 | 0 |
| | $S0$ | 0 | 0 | 0 | 0 | 0 | 0 | 0 | 0 | 0 | 0 | 0 | 0 | 0 | 1 | 1 | 0 | 0 | 1 |
| IR Agreement | | 0.888 | | | 0.776 | | | 0.944 | | | 0.832 | | | 0.469 | | | 0.776 | | |

(P5) is the one with lower scores assigned, which is in line with the numerical results that were discussed in the previous section. While most of the parameters met substantial or almost perfect inter-rater agreement, the amount of false positives only met moderate agreement.

## 5 Conclusion

A new method based on the $UNet$ architecture was proposed for the segmentation of the mitral valve leaflets. The architecture was parameterized and trained for each one of the target echocardiographic

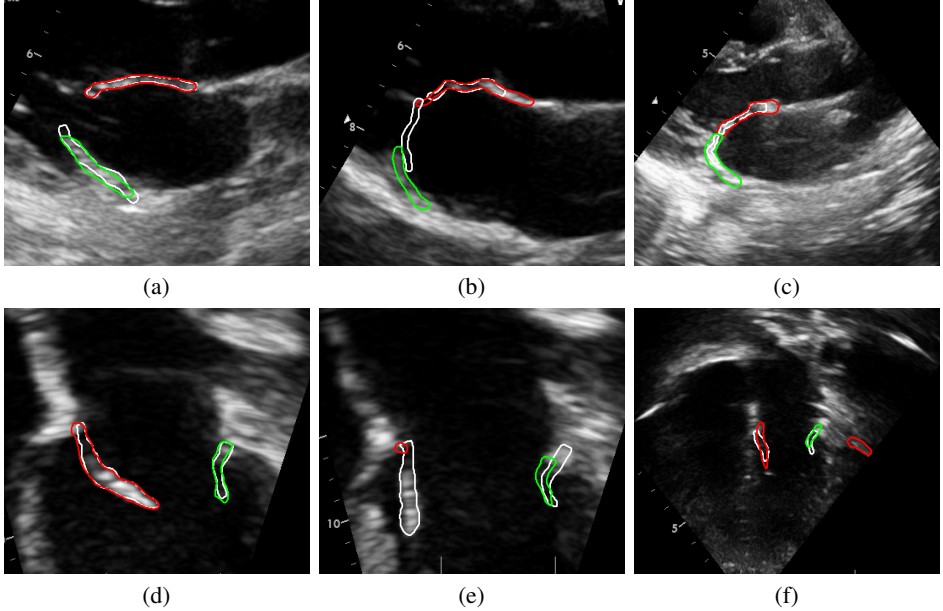

Figure 5: Example of segmentation results. White contours correspond to manual annotations and red-green to AMVL and PMVL automatic segmentations respectively. $1^{st}$ row: best (a) and worst (b) results for PLAX view; (c) is an example of error by overestimation. $2^{nd}$ row: best (d) and worst (e) results for A4C view; (f) is an example of false detection.

views, resulting in two models. Results show that both models perform in a similar way, with slight superior performance in the $A4C$ model. Moreover, they indicate a median DICE coefficient of 0.74 in $PLAX$ and 0.79 in $A4C$ for the anterior mitral leaflet segmentation, while median DICE of 0.60 in $PLAX$ and 0.69 $A4C$ are met for the posterior leaflet. By analyzing the recall and precision scores it is possible to understand that the most significant source of error is the false detection. Visual inspection of the results allows to identify two kinds of false detection: overestimation of the structures' borders and false structures detection caused by imaging artifacts. Future developments should include application of post-processing techniques, which may have a significant impact on the false positives elimination.

The clinical evaluation of the segmentation results is in agreement with the quantitative results. The parameter with the lowest scores is the amount of false positives, although the agreement is only moderate enforcing the challenge of this task.

In the future, further model optimization should be tested, as well as include data augmentation to simulate different acquisition settings. The database should also be expanded with representative examples. The clinical evaluation of the results should be continued with more cases to assess real world applicability of the proposed method.

**Acknowledgments**

This article is a result of the project (NORTE-01-0247-FEDER-003507-RHDecho), co-funded by Norte Portugal Regional Operational Program (NORTE 2020), under the PORTUGAL 2020 Partnership Agreement, through the European Regional Development Fund (ERDF). This work also had the collaboration of the Fundação para a Ciência a e Tecnologia (FCT) grant no: PD/BD/105761/2014 and has contributions from the project NanoSTIMA, NORTE-01-0145-FEDER-000016, supported by Norte Portugal Regional Operational Program (NORTE 2020), through Portugal 2020 and the European Regional Development Fund (ERDF). GE and PHILLIPS for providing the equipment. Health professionals from Círculo do Coração for their volunteer work and data collection. The Health Secretary of Paraíba for their support to the actualization of the Heart Caravan.

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
