# OpenReview forum: "Mitral Valve Leaflets Segmentation in Echocardiography using Convolutional Neural Networks"
_MIDL.amsterdam/2018/Conference — Submitted to MIDL 2018_

### Review · AnonReviewer1 · 2018-05-04
**UNet-based segmentation for echocardiography**

**Rating:** 2
**Confidence:** 2

**Review:**

The paper describes an automatic segmentation approach of the mitral leaflets. The solution is based on a well known UNet architecture and it is evaluated on two datasets: PLAX and A4C.

Structure of the introduction:
	The section reviews nicely motivation and state-of-the-art; however, the introduction of the paper lacks mention about what is the contribution of the paper.

Methodology:
	From Section 2, it seems that there is no methodological contribution. The paper uses UNet and adds zero padding in the convolutions as well as batch norm before concat operation. This modifications are rather minor and the effect of these modifications is not analyzed in the paper. Given this, it is hard to justify the the statement form the conclusion section: "A new method based on UNet architecture was proposed...".

	"This adds regularization effect by ensuring that the concatenated feature maps have the same order of magnitude". It is not clear why the regularization would be triggered by ensuring same order of magnitude. Could the authors comment on this?

Experimental section:
	Is there any data augmentation used? What is the setup of data pre-processing?

	Although the UNet model is a good choice of baseline, there is no comparison to alternative architectures nor approaches making it hard to evaluate the results presented in the paper.

**Special Issue:**

No

---

> ### Comment · ~Eva_Costa1 · 2018-05-15
> **Response to AnonReviewer1**
>
> We thank you for your review and comments about our work. Please find below our clarification to the comments.
>
> We use batch normalization layers in order to maintain filter weights in the same order of magnitude, so that they will not take large and discrepant values, preventing overfitting.
>
> Experimental section:
> Data augmentation is not used. As stated in the Conclusion section, in future work we would like to evaluate the effect of data augmentation on simulating different ultrasound acquisition settings.
> The images are not pre-processed, they are only trimmed to the dimensions of 416 x 512 pixels.
>
> We agree that our work is a baseline from which we can develop in the future and that is what we aim to do. Our main purpose was to tackle a medical problem which remains unsolved by the traditional non-ANN approaches in the literature.

---

### Review · AnonReviewer3 · 2018-05-07
**Straight-forward application of U-net for interesting task**

**Rating:** 3
**Confidence:** 3

**Review:**

Straight-forward application of a U-net to the task of segmenting mitral valve leaflets (anterior + posterior) in echocardiography. The paper has a solid overview of classical (pre-deep learning) approaches to this problem. No prior use of deep learning on this topic has been reported. The dataset is interesting, but apparently proprietary (~50 US videos from two manufacturers with more than 2500 frames for PLAX and A4C views each, 27 / 28 of which have two manually annotated mitral leaflets). The evaluation is based on voxel-wise classification measures plus subjective ratings on 23 unlabeled videos. The performance on the labeled test set looks quite good for the anterior leaflet (DSC around 0.75 can be considered a success for such a thin structure), while the posterior leaflet, which is also hard to make out for human experts, cannot be reliably segmented in all cases.
"The samples were randomly shuffled and divided into a training and validation set": I wonder if the random shuffling was done with care, or if different US frames from the same patient may have been in the training & validation sets.  Furthermore, normally an independent test set is also used for deep learning.  Update: Later, Table 1 reveals that there is a test set, but it should be addressed also in section 2. Furthermore, it looks like validation & training sets are not independent.
The authors should really try to understand the overlapping tile strategy (see e.g. the original Ronneberger U-net paper [13]) which does not only mean to use "valid" convolutions instead of zero padding, but also allows to process arbitrarily large images ("Due to memory management and taking into account the down-sampling process, samples’ dimensions were set to a 416 pixel height and 512 pixel width."). The training speed-up that "was not observed during preliminary tests" is due to the fully-convolutional nature and the ability to freely choose the training patch sizes, and cannot be observed of course when comparing training with small patches and zero padding against training with much larger, padded inputs.
The number of feature channels was quite low: 4 to 32 in the first layer (but Table 2 only contains results for 4, 8, 16, which surprises the reader before reading the next section), whereas the original paper suggested 64, and may not be doubled in every level, although that looks like an error in Fig. 2. (Note that Cicek et al. proposed a refined scheme in 2016 for 3D to save memory and prevent bottlenecks, which could also have been applied here, but was not.) That is probably because of the limited GPU (GTX 970 with 4 GiB memory), which I do not want to criticize, but could have been made better use of.
The fact that less filters worked better for PLAX (8 instead of 16) could be due to missing regularization / dropout. Furthermore, no data augmentation is mentioned, although it is commonly done with U-Nets and would likely improve the models (when chosing appropriate transformations, of course).
Overall, the paper structure could be improved a little bit; often things appeared to be missing, but would be mentioned later. There are some technical shortcomings (see above, plus some obvious options for improvements already mentioned by the authors), but they are not major, and the topic is interesting and has not been addressed before.
Small comments:
"… the DICE, precision and recall were used.", but that should be further defined. Are they defined on the voxel level? Aggregated per frame? or per batch? per object?
- "greed search optimization": probably "grid search"?
- "Preliminary tests have shown that lower depths result in less accurate results." does not belong into 2.1.1, but the results section.
- strange formatting of abbreviations (PLAX in LaTeX math mode? looks like P*L*A*X this way) and some more words (CNN, Sscore, …)
- "a batch size of 4 was set" → use unbreakable space before 4

**Special Issue:**

No

---

> ### Comment · ~Eva_Costa1 · 2018-05-15
> **Response to AnonReviewer3**
>
> We thank you for your review and comments about our work. Please find below our clarification to the comments.
>
> Concerning the training and validation sets, due to the limited data available, they are not independent, so different frames from the same patient may be included in training and validation. We are aware of the risk of overfitting. However, our test dataset is completely independent of the train and validation dataset and the results are similar to those observed in the validation. In the future, with a bigger and more diverse dataset, we should use independent sets.
>
> Fig. 2 features a typing error in D=3, where should be N3=2N2 instead of N2=2N1. The number of layers is doubled at every level. We are aware that the number of layers is limited, but we have reached hardware exhaustion. We thank you for your suggestions about the GPU usage and we’ll consider them in the future.
>
> Concerning data augmentation, we mention in the Conclusion section that in a future work we want to test its effect by simulating different US acquisition parameters.
>
> Small comments:
> The DICE values for validation (Subsection 4.1) were computed pixel-wise for each frame and averaged over all classes and frames in the batch.
> DICE, Precision and Recall for testing (Subsection 4.2) were computed pixel-wise for each frame and for each by class.

---

### Review · AnonReviewer2 · 2018-05-08
**Unet application**

**Rating:** 1
**Confidence:** 2

**Review:**

The authors evaluate Unet for an automated localization and segmentation of mitral valve leavelet segmentation in echocardiographic images.

Pro: This is an important application, and results are promising. Evaluation seems to be well designed and thorough. Unet performs better than previous non-ANN applications.

Con: While the study features an interesting application and data set, there is little algorithmic novelty. It features little more than an application use case for Unet.

**Special Issue:**

No

---

> ### Comment · ~Eva_Costa1 · 2018-05-15
> **Response to AnonReviewer2**
>
> We thank you for your review and comments about our work.
> We agree there is little algorithmic novelty in our work. The focus was on tackling a medical problem which remains unsolved by the traditional non-ANN approaches in the literature. We believe that our contributions are the creation of a baseline from which better solution can emerge and provide a medical perception of the results, even when they are not numerically great. Furthermore, we believe our work could be of interest in a more applicational session.

---

### Decision · Program_Chairs · 2018-05-15
**Paper34 Acceptance Decision**

Reject